# The Effect of Climate on Strongly Disturbed Vegetation of Bait Sites in a Central European Lower Montane Zone, Hungary

Katalin Rusvai [1], Judit Házi [2,*] and Szilárd Czóbel [3]

1   Department of Nature Conservation and Landscape Management, Hungarian University of Agriculture and Life Sciences, Páter Károly utca 1., H-2100 Gödöllő, Hungary; kissne.rusvai.katalin@uni-mate.hu
2   Department of Botany, University of Veterinary Medicine Budapest, Rottenbiller utca 50., H-1077 Budapest, Hungary
3   Institute of Plant Sciences and Environmental Protection, Faculty of Agriculture, University of Szeged, Andrássy út 15., H-6800 Hódmezővásárhely, Hungary; czobel.szilard.endre@szte.hu
*   Correspondence: hazi.judit@univet.hu

**Abstract:** Human landscape-transforming activities contribute to the global change in vegetation in different forms. Hunting is one of the most ancient human landscape-shaping activities. Feeders for hunting are particularly disruptive to vegetation. In the present study, we conducted a vegetation survey in these highly disturbed places. We investigated the vegetation dynamics over several years in the turkey oak–sessile oak zone, in two areas with different moisture and shade conditions (forest and clearing). Important background factors are the changes in precipitation and temperature. Our results confirm that weed infestation is detectable at bait sites over a long period. The seasonal changes in field weed vegetation, as well as the increase in the number and coverage of weed species at the end of summer, resulting from lifestyle characteristics, were generally detectable in all years and locations, especially in the case of open and more strongly degraded clearings. Meteorological factors played a role in the degree of weed infestation in each year. Degradation was more significant in drought years, while regeneration was also observed in wetter periods. At baits located in the clearing, we showed a positive correlation between the amount of summer precipitation and the total coverage of weed species, as well as between the average spring temperature and the coverage of certain weed species. With the drying of the climate, the disturbed areas are constantly losing their natural value, but wetter weather is not an automatic solution either. Considering that there are approx. 30,000 bait sites in the country, and they are used regularly and very intensively, they can serve as major infection hotspots for alien species in a network.

**Keywords:** bait site; wild boar hunting; degradation; climate change; plant invasion

## 1. Introduction

Feeding wildlife is a controversial method with diverse effects on ecosystems, animal welfare, and public safety. Despite the level of baiting and the supplementary feeding of wild game, our understanding of the ecological effects of these types of food support is still very limited [1]. Artificial feeding may have direct benefits, like reducing mortality, enhancing the physical wellbeing of game, maintaining body weight, and reproductive performance [2]. However, it can lead to severe problems, like changes in social behavior, territorial behavior, and the natural migration and activity patterns of animal species; it can cause an increase in competitiveness, lead to negative interactions between animals, and increase the risk of disease transmission [3,4].

The effects of wild game feeding have already been widely investigated, but mostly the animal species have been the focus, and mainly the effects of supplementary winter feeding were examined in northern countries [2,3,5]. However, in Central and Eastern Europe feeding for hunting purposes is the most widespread type of food support [6]. At these sites, the goal is primarily to bait wild boar and thereby facilitate the hunt. A bait

site is a small clearing established approx. 30–50 m from hunting blinds, where feed is generally scattered on the ground [1]. Usually corn-cobs or seeds are used at the sites; although, agricultural and food industry by-products (e.g., molasses, fresh and dried beet slices) are also applied [1]. Given that agricultural products, especially cereals, contain weed seeds [7,8], this practice can lead to the invasion of natural habitats by noxious weeds and exotic species. Due to the disturbance occasioned by extended anthropogenic activity and increased game density, the bare and degraded soil produced by trampling, and the greater availability of nutrients, all bait sites may be considered important potential aids to invasion [9,10].

In Hungary, bait sites for wild boar hunting are very widespread, and the problem is not only their large number (according to estimates, around 30,000 are in operation across the country) but also their extremely intensive use. According to the wildlife management database [11], an average of nearly 150,000 tons of fodder is dumped on these sites every year, which significantly endangers the natural habitats due to fodder contaminated with weed seeds and constant vigorous disturbance. Nonetheless, only in recent years has more attention been paid to this problem. Significant degradation at bait sites was showed in different habitat types in Hungary [12,13] and also in Slovakia [14]. It has been proven that the operation of bait sites over several years can significantly damage the physical and chemical parameters of the soil and degrade the soil seed bank as well [13]. It has also been proven that in case of abandonment of these places, the abundance of weeds decreased significantly over time, but the number of weed species typically did not change, and the presence of arable weed species was detectable even after almost a decade [15]. Moreover, the density of weed seeds in the soil seed bank can also remain significant even after 10 years of abandonment [15]. In Slovakia, the number of non-native plant species in the national parks is increasing every year, and some researchers mention that these feeding grounds are one of the main sources of these plants [16]. Others found that some exotic species can be detectable even in the mountainous environment at higher and higher altitudes, even above 1000 m near these sites [14]. Therefore, it may be that climate change can also affect the presence and the abundance of weed species at the feeding grounds and, accordingly, it can affect the degree of degradation and possible regeneration processes at these places.

The aim of this study is to examine the impacts of baiting on herbaceous vegetation at bait sites in a typical Central European lower montane zone. We investigated the vegetation dynamics in two vegetation zones, oak and beech zones, in two areas with different moisture and shade conditions (forest and clearing). The main questions of the study were the following:

1. How does the extent of weed coverage change over the years and whether seasonal changes are detected?
2. How does precipitation and temperature affect the number and the coverage of weed species in each year?

## 2. Materials and Methods

### 2.1. Description of the Study Area

Altogether 6 bait sites were included in the assessment (Figure 1). Three forest (F1, F2, F3) and three clearing sites (C1, C2, C3) were investigated in 2016, 2018, 2019, and 2020 in a Central European Lower Montane Zone, in Hungary, in the turkey oak–sessile oak (*Quercus cerris* L.–*Q. petrea* (Matt.) Liebl.) zone. Since there is no official public register of the bait sites available, we have chosen those that have certainly been in use for at least 5–10 years and are located in the forest or clearing (not in edge habitat). The forest sites were situated in oak stands, where the canopy cover was at least 80%, and the forest floor was characterized by dense litter cover and sparse undergrowth. Clearing sites refer to small patches (50–100 m in diameter) of dry, mesophilous mountain meadows, which were created by tree cutting several decades ago. The selected sites are located on flat-topped

ridges or saddle areas with gently sloping and slight southern exposure at an altitude of 400–500 m above sea level.

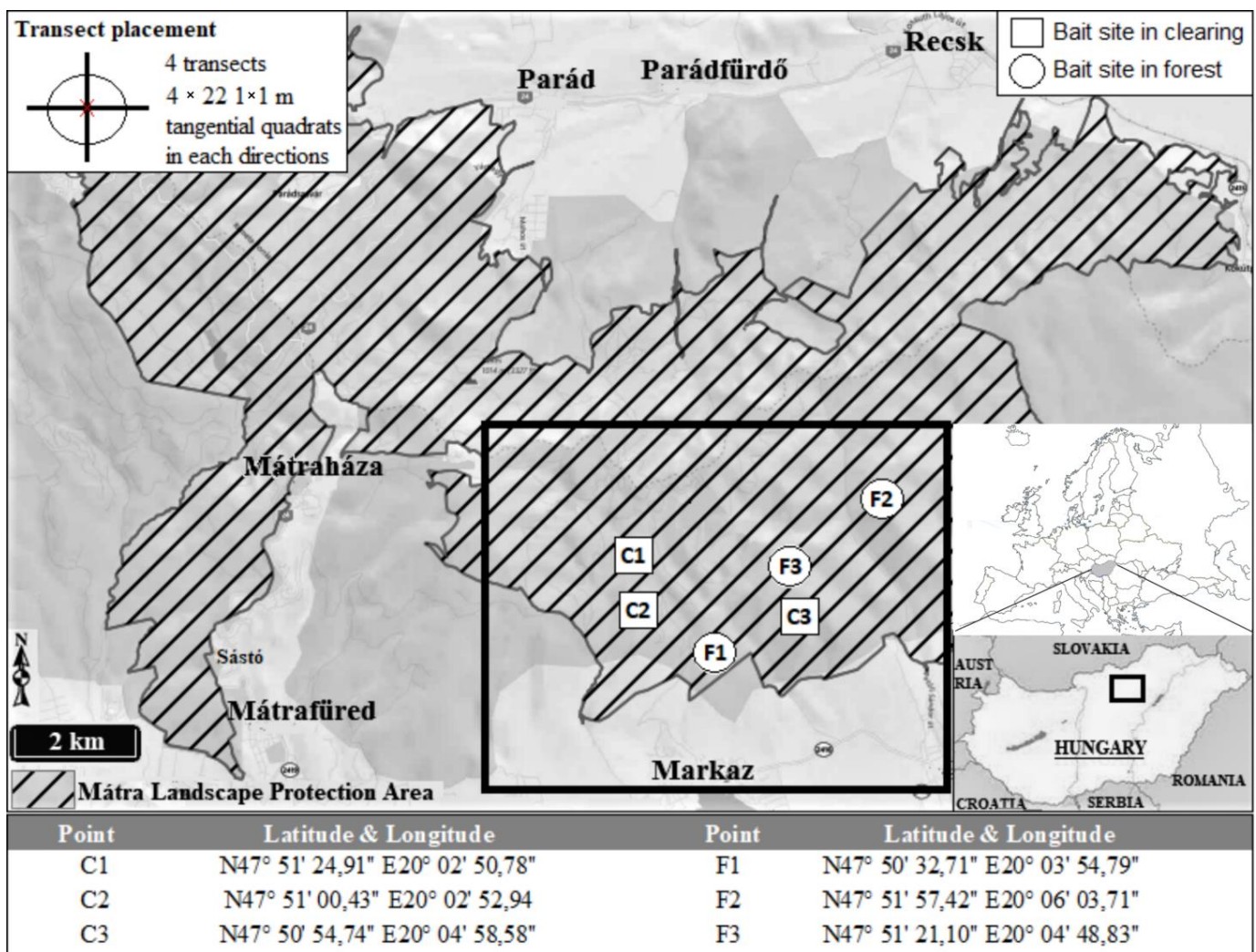

**Figure 1.** Arrangement of the survey with the map and GPS coordinates of the sampling points.

At all places, vegetation was investigated using the transect method. The transects were set out from the center of the baits in 4 directions, closing an angle of 90° to each other. A total of 22 tangential quadrats (1 × 1 m) were placed on each of them, in which a percentage coverage estimation was carried out in May and August of each year. Periods were chosen to examine whether the well-known interannual changes in the arable fields [17] can also be detected at bait sites. In the present study, the coenological data of the individual units are evaluated together. To evaluate the role of the meteorological parameters, average daily temperature and daily precipitation data between 2016 and 2020 were used. Data were collected from the nearest weather station of the National Meteorological Network, using the Meteorological Database [18].

### 2.2. Coenosystematic Classification

The vegetation of the bait sites was evaluated according to type (forest vs. clearing) and examination period (May and August), based on their species list and abundance (cumulative coverage of the taxons per bait site). To estimate the naturalness of communities, Borhidi Social Behaviour Type (SBT) classification [19] was used. This is an adapted version of Ellenberg's grouping [20] and Grime's competitor/stress-tolerator/ruderal (CSR) plant functional type system [21] to Pannonian flora. This grouping derives from species

behavior and ecological attributes at a given observation level [19]. Social behavior can be defined as the role that a plant species plays in the community, considering its ecological characteristics, morphology, and physiological performance. According to this grouping, the following two main groups were used: (1) Naturalness indicator species (natural species): stress tolerant specialists—S; competitors of natural habitats—C; stress tolerant generalists—G; natural pioneers—NP; disturbance tolerant plants—DT. (2) Degradation indicator species (weed species): native weed species—W; introduced crops running wild—I; adventitious weeds—A; ruderal competitors of the natural flora—RC; alien competitors, aggressive invaders—AC. The naturalness of the communities was evaluated based on the proportion of these groups. The ratio of the species groups, as well as their proportion of the cumulative coverage was also calculated. To examine the role of meteorological factors, we calculated monthly averages obtained from the basic data and compared them with the cumulative coverage of weeds and natural species. The number and the abundance of alien plant species was also calculated to evaluate the role of bait sites in the maintenance of these plants. Species names were provided based on the nomenclature of Király [22].

*2.3. Statistical Analysis*

The effects of different vegetation environments of different sites, the consecutive years on total coverage, and the number of species of different indicators were compared using analyses of variance (ANOVA) with the sites and years as fixed factors.

During the analysis of variance (ANOVA), we examine the differences between and within the groups; that is, we divide each sum of squares by the corresponding degrees of freedom. The result is actually a measure of variance. The F-ratio is then obtained by dividing MS (between) and MS (within). And the *p*-value is the probability of obtaining an F-ratio greater than the observed one, assuming that the null hypothesis that there is no difference between the group means is true. For a post hoc test, the Tukey Honestly Significant Difference (HSD) with corrections (adjusted *p*-values for the multiple tests) was used.

The analyses were performed with R statistical software (version 4.0.5, R Core Team 2021 [23]) using the packages "tidyverse" and "rstatix". For linear regression, we used the PAST program [24]. We used a comprehensive yet easy-to-use software package long used in paleontology to perform the quantitative standard numerical analyses and operations. Most of the most frequent functions used in PAST (PAleontological Statistics) can be used in paleontology as well as in ecology and are suitable for handling often incomplete data. These functions are not found in standard, more extensive statistical packages, but at the same time they can be used well in university education.

## 3. Results

*3.1. Species Pool and Abundance*

Altogether, 224 species were found at the six bait sites in the 4-year study period. A third of this (74 species, 33%) were weed species. The previously confirmed habitat differences [12] were clearly detectable in every year. At the clearing sites, more weed species were found in higher abundance, and the species pool was also different. A total of 181 taxa were detected at the feeding sites located in the clearing, of which 71 were degradation indicator species (39.2%), while only 134 species were present at the forest sites, of which 37 (27.6%) were weeds.

In terms of the most abundant species, the forest sites were dominated by natural weed species, like *Rumex crispus* L. and *Fallopia convolvulus* (L.) Á. Löve (Table 1). However, the coverage of these species was generally very low. The maximum coverage of *Rumex crispus* L. per quadrat was only 65%, while the average coverage of the other degradation indicator species was typically only 10–20% (*Chenopodium album* L.—20%; *Fallopia convolvulus* (L.) Á. Löve—15%; *Plantago major* L.—15%; *Galium aparine* L.—10%; *Ambrosia artemisiifolia* L.—10%; *Datura stramonium* L.—10%). The naturalness indicator species proved to be more abundant in all forest sites. At greater distances from the center of the baits, the coverage of these species was about 60–80%. Only *Urtica dioica* L. was able to reach 100% in some

quadrats, mainly those close to the feeding places. However, the mass of disturbance-tolerant species (DT species) is still significant. Despite all of this, even within the group of natural species, disturbance-tolerant species (Borhidi indicator 'DT' category) dominated, which, in addition to the low plant coverage values, clearly indicates the disturbance of these locations.

**Table 1.** The 10 most abundant naturalness and degradation indicator species (in order) at the forest bait sites based on the 4-year investigation.

| Naturalness Indicator Species | Indicator | Degradation Indicator Species | Indicator |
|---|---|---|---|
| *Urtica dioica* L. | DT | *Rumex crispus* L. | W |
| *Carex divulsa* Stokes. | DT | *Fallopia convolvulus* (L.) Á. Löve | W |
| *Poa nemoralis* L. | C | *Chenopodium album* L. | RC |
| *Geum urbanum* L. | DT | *Plantago major* L. | W |
| *Moehringia trinervia* (L.) Clairv. | DT | *Polygonum aviculare* L. | RC |
| *Poa angustifolia* L. | DT | *Galium aparine* L. | W |
| *Festuca heterophylla* Lam. | C | *Ambrosia artemisiifolia* L. | AC |
| *Melica uniflora* Retz. | C | *Veronica hederifolia* L. | W |
| *Lysimachia nummularia* L. | DT | *Datura stramonium* L. | W |
| *Dactylis glomerata* L. | DT | *Convolvulus arvensis* L. | RC |

Notes: Naturalness indicator species (natural species): C—competitors of natural habitats; G—stress tolerant generalists; DT—disturbance tolerant plants. Degradation indicator species (weed species): W—native weed species; RC—ruderal competitors of the natural flora; AC—aggressive invaders.

At the clearing sites, the species pool and density were completely different (Table 2). Ruderal and segetal weeds proved to be the most abundant in all three locations, and the average coverage values of these species in the sampling units were also higher than in the forest sites. *Polygonum aviculare* L., *Xanthium spinosum* L., *Datura stramonium* L., and *Tripleurospermum inodorum* (L.) Sch.Bip often reached 100% coverage in the quadrats, and other species could also very abundant (e.g., *Bromus sterilis* L.—95%; *Capsella bursa-pastoris* (L.) Medik—90%; *Ballota nigra* L.—90%). Moreover, at these sites, alien species were also able to represent a significant mass. For example, *Ambrosia artemisiifolia* L. and *Echinochloa crus-galli* could reach 15–20%, while other non-indigenous plants, like *Galinsoga parviflora* Cav. and *Erigeron annuus* (L.) Pers. reached a maximum of 5%.

**Table 2.** The 10 most abundant naturalness and degradation indicator species (in order) at clearing sites based on the 4-year investigation.

| Naturalness Indicator Species | Indicator | Degradation Indicator Species | Indicator |
|---|---|---|---|
| *Poa angustifolia* L. | DT | *Polygonum aviculare* L. | RC |
| *Carex praecox* Schreb. | G | *Xanthium spinosum* L. | W |
| *Fragaria viridis* Duch. | G | *Tripleurospermum inodorum* (L.) Sch.Bip | W |
| *Festuca rubra* L. | C | *Bromus sterilis* L. | RC |
| *Achillea collina* J. Beck | DT | *Datura stramonium* L. | W |
| *Bromus hordeaceus* L. | DT | *Capsella bursa-pastoris* (L.) Medik | W |
| *Euphorbia cyparissias* L. | DT | *Convolvulus arvensis* L. | RC |
| *Festuca pseudovina* Hack. ex Wiesb. | C | *Ballota nigra* L. | W |
| *Urtica dioica* L. | DT | *Chenopodium album* L. | RC |
| *Galium verum* L. | DT | *Rumex crispus* L. | W |

Notes: Naturalness indicator species (natural species): C—competitors of natural habitats; G—stress tolerant generalists; DT—disturbance tolerant plants. Degradation indicator species (weed species): W—native weed species; RC—ruderal competitors of the natural flora.

In terms of the spatial distribution of the above-named species along the transects, the previously confirmed stress gradient [12] was also proven to be detectable in every year, every sampling period. The above-mentioned degradation indicator species were the most abundant in the center of the bait sites, especially in clearings, where the presence

of weed species was dominant until 5–6 m from the feeding sites. Meanwhile, the density of natural species increased with the distance. It should be emphasized, however, that although the continuous weed coverage did not usually spread further than 5–10 m, the presence of weed species could be detected even at greater distances in every year and every examination period, which clearly indicates the long-term degrading effect of these hunting facilities.

We started the exploration of the data structure by separating the number of species and the coverage values of the two large areas of the forest and the clearing. In general, it can be stated that there is a big difference between the samples in the forest and in the clearing in terms of cumulative coverage of naturalness and degradation (Figure 2) indicator species. According to the ANOVA, the number of native and also degradation indicator species was significantly higher ($p < 0.001$) in the clearing sites.

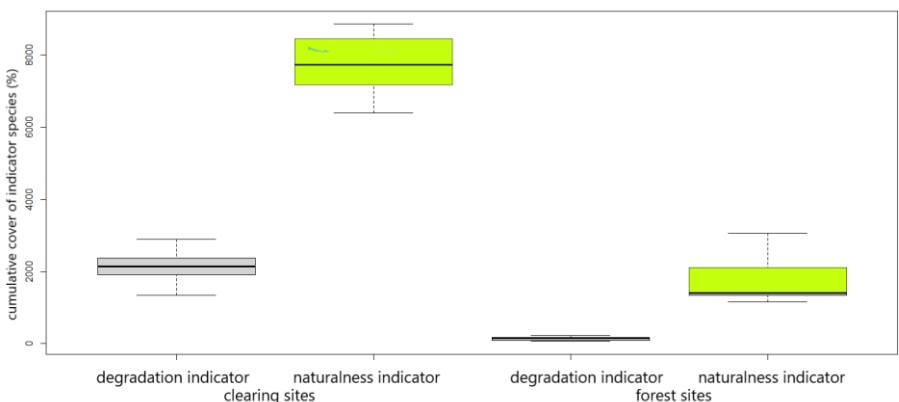

**Figure 2.** The cumulative coverage of naturalness indicator species and degradation indicator species on the forest and clearing sites.

Comparing the total coverage value of naturalness indicator species between different sites, significant differences were detected only between the clearing and forest sites. (Figure 3) (F = 14.515, $p = 3.172 \times 10^{-9}$ ***, Df = 7). For the results of consecutive years, there was no significant difference in the coverage results, even between the initial and the last year of the study.

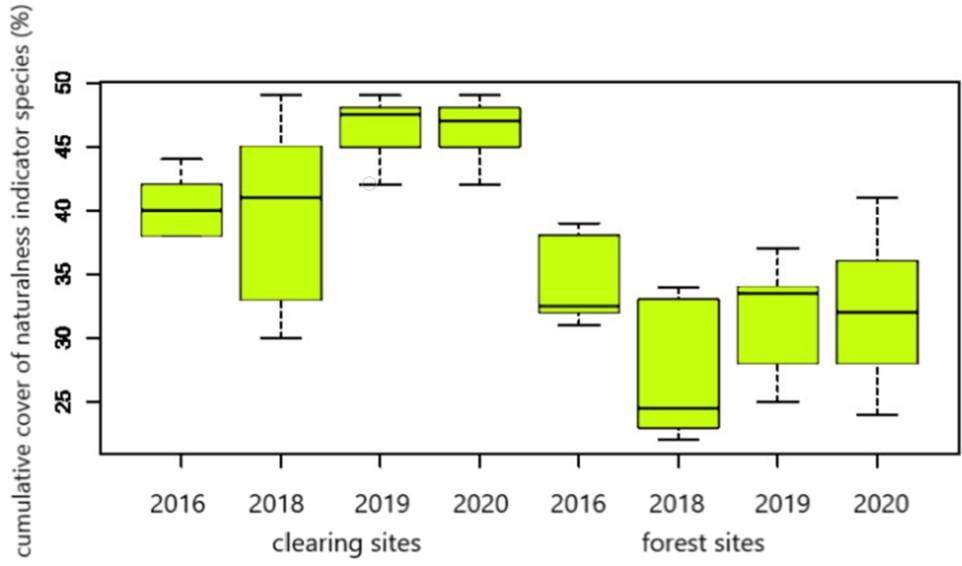

**Figure 3.** The cumulative coverage of naturalness indicator species on the forest and clearing sites in consecutive years.

Comparing the total coverage value of degradation indicator species between different sites, significant differences were detected only between the clearing and forest sites. (Figure 4) (F = 15.797, $p$ = 9.794 × 10$^{-10}$ ***, Df = 7). For the results of consecutive years in this case, a clearing site showed significant difference in the coverage results, between 2016 and 2018 and the last year of the study ($p$ = 0.0012628, $p$ = 0.0305688, respectively).

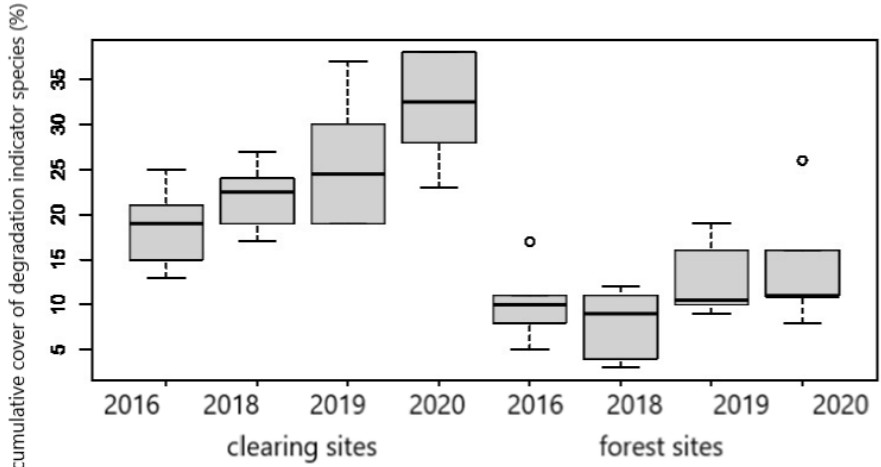

**Figure 4.** The cumulative coverage of degradation indicator species on the forest and clearing sites in consecutive years. The circles represent outliers.

*3.2. Intra- and Interannual Changes in Vegetation*

The well-known seasonal changes in field weed vegetation [17] proved to be true in the case of bait sites: the increase in the number and coverage of weed species at the end of summer, resulting from their life-form, was generally detectable in all years and locations (Figure 5).

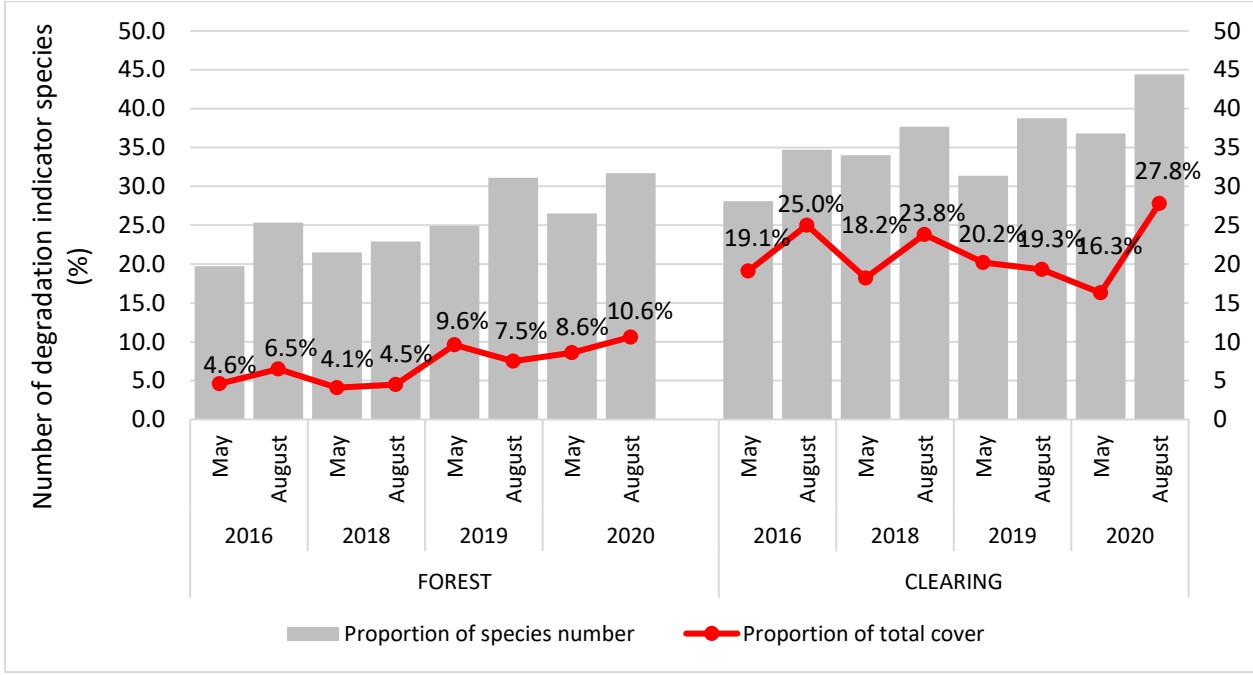

**Figure 5.** The cumulative coverage of degradation and naturalness indicator species in the average of bait sites located in the forest sites and in the clearing sites in May and August 2016–2020.

The difference between the two habitat types is also clearly visible in this Figure, according to which the bait sites located in a clearing turned out to be much more degraded.

Based on the 4-year recording data, the total number of species, the number of degraded species, and the number of natural species were significantly higher ($p < 0.001$) in baits located in the clearing, and the cumulative total coverage of each species was also much higher in these locations. However, the rate of growth over the years (and also within individual locations) was very variable, and an increasing trend was also observed in the mentioned parameters during the study, the reason for which can presumably be revealed by examining the meteorological and other parameters.

*3.3. Influence of Meteorological Factors on Vegetation*

The extent of weed infestation was significantly influenced by the examined meteorological factors. At the clearing sites, positive correlation ($R^2 = 0.975$) between the amount of summer precipitation and the total coverage of degradation indicator species was shown, as well as the average spring temperature and the coverage of naturalness indicator species ($R^2 = 0.831$) (Figure 5).

Meanwhile, the distribution of precipitation also proved to be a decisive factor: in particular, warm, dry springs, and the following rainy summers increased weed density (Figure 6). In 2016 and 2020, there was a very dry spring followed by a relatively wet summer, as a result of which, after the sparse weeding in May, the mass of large weed species increased significantly by August. In the years 2018 and 2019, with more spring precipitation, the changes between the two examination periods were generally not that significant. At that time, the weed species already appeared in May, so the rate of growth in these years was not so significant. Moreover, in the year 2019, when both the spring and summer were especially rainy, the coverage of weed species even decreased slightly in terms of average values, while the abundance of natural species increased (Figure 7).

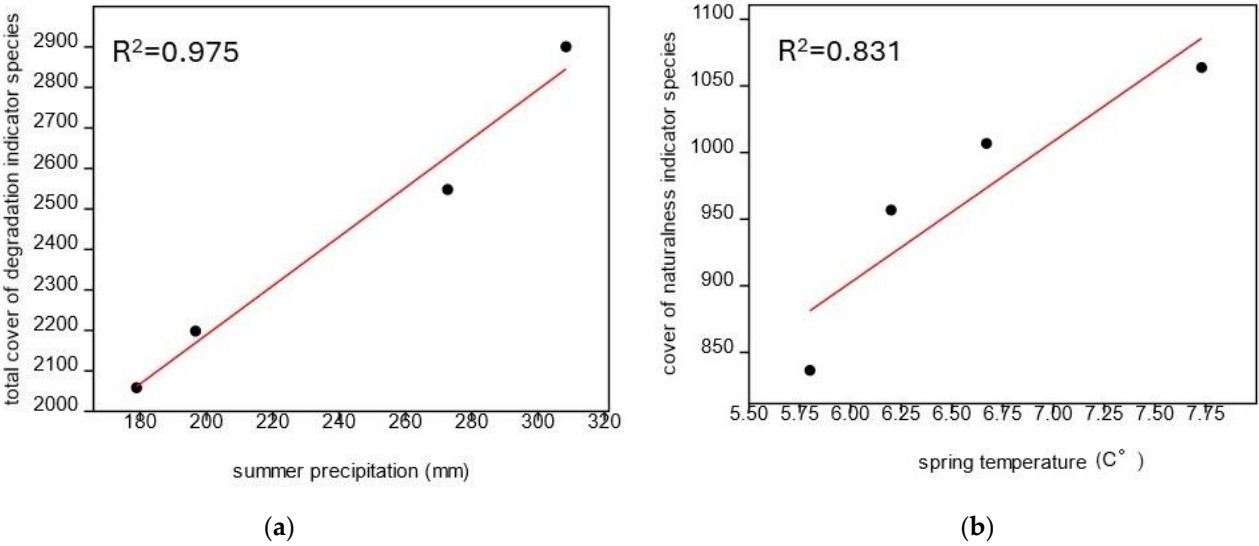

(**a**)　　　　　　　　　　　　　　　　　　　　　　　(**b**)

**Figure 6.** Correlation between some meteorological factors and weed abundance in the case of bait sites located in a clearing. (**a**) Correlation between the amount of summer precipitation and the total coverage of weed species. (**b**) Correlation between the average spring temperature and the total coverage of native weed species.

Presumably temperature also played an important role in the characteristics of weed coverage. In springs with a higher average temperature and a relatively large amount of precipitation (in years 2018 and 2019), significant weed growth could already be developed in May. The most characteristic example is the case of *Xanthium spinosum* L. appearing in one of the sites. This species had already appeared in high density by May in 2018. This is due not only to the relatively large amount of spring precipitation but also to the fact that this year was the warmest spring (average temperature: 7.7 °C). And since this weed

requires a particularly high soil temperature for the initiation of germination, in the colder springs of 2019 and 2020 (average temperature: 6.2 °C and 5.8 °C), this species could only appear in August, and typically not in too large density either.

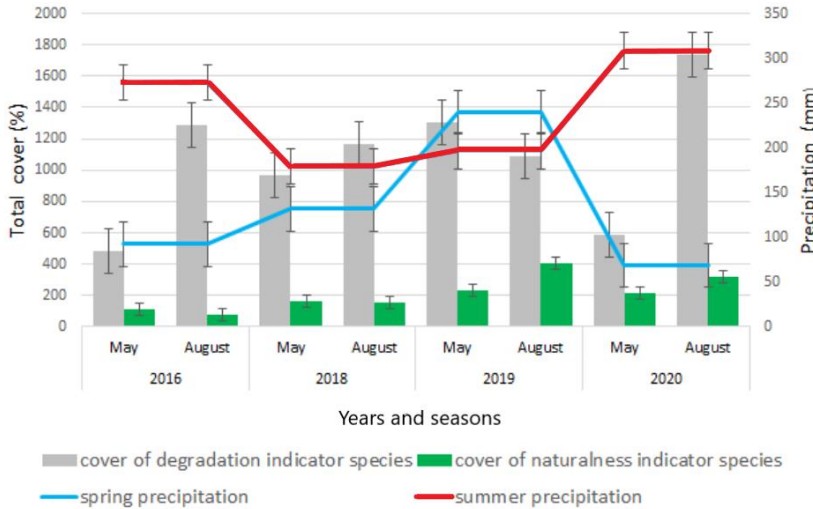

**Figure 7.** Cumulative total coverage (∑%) of degradation and naturalness indicator species in relation to spring and summer precipitation (mm) and average temperatures (°C) in the center of the bait sites located in a clearing (1–5. quadrats), considering the average of the bait sites in each study period.

Considering the abundance of alien species, it can be said that the coverage of non-native plants was very variable in each year and in each location. The proportion of these species from the cumulative weed coverage varied between 2.2% and 46.5% during the study periods. However, if we examine the above-mentioned meteorological factors, it can be seen that the interannual changes within the year significantly affect the abundance of these species. The coverage of alien plants increased to a much greater extent in the years (in 2016 and 2020) when a dry spring was followed by a rainy summer (Table 3).

**Table 3.** Interannual changes (May to August) in cumulative coverage of weed species at bait sites located in clearing.

|  | 2016 | 2018 | 2019 | 2020 |
|---|---|---|---|---|
| Alien species | +790% | +46% | +514% | +1763% |
| Native weeds | +3% | +1% | −10% | +78% |
| Total weed coverage | +21% | +18% | +0.4% | +116% |

Note: Values show percentage increase in August, compared to May, based on the cumulative coverage of the listed species groups. (Gray areas represent the most significant interannual changes).

Although invasive alien plant species (IAS; according to Lambdon et al. 2008 [25]) usually account for only a small proportion of the total weed coverage (0.2 to 3.6%), their presence is definitely important, especially because their coverage has increased over the years (Figure 8). The most abundant species were *Ambrosia artemisiifolia* L., *Conyza canadensis* L., and *Erigeron annuus* (L.) Pers., which are all alien species spreading throughout Europe [25]. Neither the temperature nor the precipitation data could support the trend, so it can presumably be the result of several other factors.

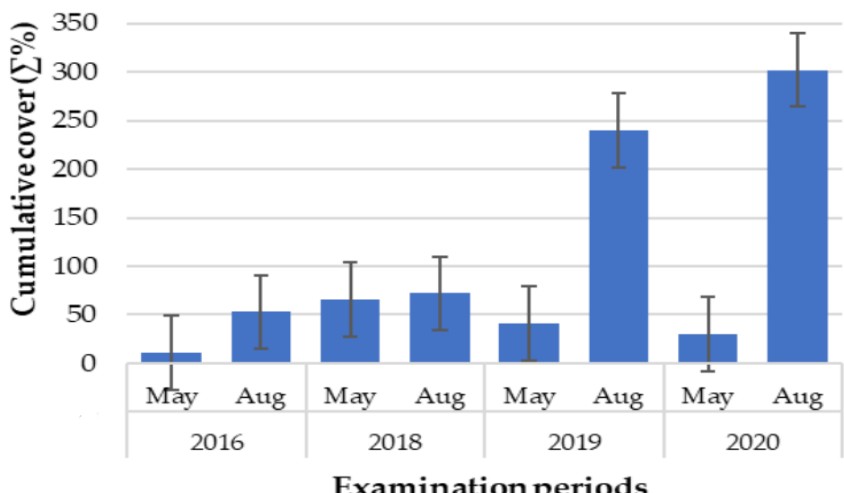

**Figure 8.** Changes in the abundance of invasive alien plant species (IAPS) during the 4-year study at the baits sites located in clearing.

## 4. Discussion

### 4.1. Species Pool and Abundance

Based on the 4 years of investigation, it has been proven that feeding can cause significant and long-term degradation in natural habitats. Our results are compatible with those experienced in Slovakia, in connection with hunting establishments. Thanks to the repeated supply of diaspores, the presence of species of external origin remains local, but constant [14]. However, the extent of this was different in the two habitat types. Thanks to the different moisture and shade conditions, it was shown that feeding places located in the clearing are proven to be much more degraded. More weed species were found in higher abundance, and mainly ruderal and segetal weeds dominated. Meanwhile, in forest sites, the vegetation was sparse, and it was characterized by natural species, appearing mostly in small patches. The previously proven stress gradient [12] was also consistent with this habitat difference in all years. The coverage of degradation indicator species was the highest in the center of the bait; further away from the center, their density and the number of species decreased, while the number and coverage of natural species increased. However, the spatial extent of this was different for each bait type. As expected, the bait sites located in the clearing proved to be the most degraded. In this case, the area of the baits was typically characterized by continuous weed coverage until 5–8 m, while in the forest sites, the bare, litter-free soil surface was dominated by only a few weeds.

All these phenomena are presumably due to the higher sensitivity of open habitats to invasions [26] and the specific environmental needs of weed species [27]. In the clearing sites, many segetal and ruderal species were able to form homogeneous stands. As has also been shown in Slovakia [14], species such as *Datura stramonium* L., *Xanthium spinosum* L., and *Polygonum aviculare* L. were able to form high densities in the immediate vicinity of the feeding places. This result also clearly indicates that the most species-rich communities are not necessarily the most resistant to invasions [28]. And, although weed invasion typically extends to the intermediate environment of the bait sites [12,14], many examples are known when this led to the complete destruction of habitats, e.g., [29,30]. Thus, small mountain clearings are the most endangered, especially because bait sites are mainly established in these habitats.

### 4.2. Intra- and Interannual Changes in Vegetation

There was a significant difference between the vegetation of the examined periods: In August, more weed species were present at all sites, with a higher coverage. The reason could be that most segetal weed species can typically germinate at higher soil temperatures [31], so most of them could only reach a significant density by August. And,

although this is a clear phenomenon in field environments [32], according to the results of this research, it proved to also be detectable in natural environments.

Regardless of the weather components, the variation between the years may have been partially caused by several other factors, including unique habitat characteristics and other anthropogenic disturbances, like the decreasing wild boar population due to the spread of ASP [33] and the slightly reduced hunting intensity due to restrictions caused by COVID-19 [34]. These factors together could have resulted in the fact that, with decreasing disturbance, the seeds of the weed species hidden in the soil could more easily appear on the surface, resulting in greater coverage of them. Although the disturbance was slightly reduced, the nutrient content of the soils was still very high [13], which mainly helps the growth of the species. In that way, the changes in the abundance of alien species can be partly explained by this. But since we only examined the average temperature and precipitation data here, it should not be dismissed that other weather factors may have played a role in this. For example, the uneven distribution of precipitation, the increase in frequency of extreme weather events, and also the unpredictable changes in the use of feeding sites (e.g., the quantity and quality of forage, as well as the method and frequency of their placement) can all play a role in the spread of alien species.

*4.3. The Role of Meteorological Factors*

The extent of weed infestation at bait sites was significantly influenced by the examined weather factors. The degree of degradation was more significant in drought years, while some regeneration processes were also observed in wetter periods. This result is consistent with the result of Jánoska [35], who also experienced this phenomenon in a wild boar preserve, where the level of disturbance is similarly strong. The distribution of precipitation also proved to be a decisive factor. In particular, warm, dry springs and the following rainy summers increased weed density. The reason for this is that large weed species usually germinate later [31], and they require a significant amount of water for their growth [36]; therefore, they are able to suppress natural species. Moreover, since weed and alien plant species also require higher temperatures to germinate than native plants, the increasingly frequent warmer springs will be more favorable for them [37].

As has been proven, periods of drought also promoted the growth of weed species. These effects of climate change clearly promote the spread of weeds and alien species [38]. For example, the coverage of *Ambrosia artemisiifolia* L., *Conyza canadensis* L., and *Erigeron annuus* (L.) Pers. has increased over the 4-year study, which can be examined by these facts. Among them, common ragweed was the most abundant, a species that a previous study has already indicated at bait sites across the country, sometimes in a significant density [39]. The authors highlight the role of bait sites and the road network connected to them in the spread of invasive species. The importance of this species was also emphasized by a study that examined the changes in vegetation at abandoned bait sites. It has been proven that common ragweed was able to appear even after nearly a decade of abandonment, and in this case too, it became especially abundant after warm, dry springs [40].

The spread of invasive species through hunting facilities has also been confirmed in in Slovakia, where the number of non-native plant species in the national parks is increasing every year, and some researchers mention that hunting facilities as one of the main sources of alien plants [16]. What is more, thanks to climate change, segetal weeds were found at increasing altitudes above sea level along these hunting facilities [14]. This has also been proven in Hungary, where *Abutilon thepohrasti* Medik. and *Datura stramonium* L. were found above 900 m at a bait site located in a beech forest [40]. In this research, an increase in the abundance of invasive alien species over the years was demonstrated. It means that, thanks to the effects of climate change and the increasing anthropogenic disturbances, the threat to natural habitats is increasing and mountainous areas are also endangered.

Moreover, the above-mentioned effects can be strengthened by other processes, such as the consequent weakening of forest health [41], which, combined with often inadequate forest management methods and other anthropogenic effects, can lead to the opening of

forests [42]. It can further promote the spread of weeds and invasive species [43] and also cause strong degradation in more closed forest areas [44,45] and other valuable habitat patches as well [46].

## 5. Conclusions

The degree of degradation is primarily determined by habitat characteristics (forest versus clearing), but climatic factors (mainly precipitation and its distribution) have also played a significant role in the appearance and the abundance of weed species at bait sites. Because of climate change and increasing anthropogenic and other disturbances, these feeding places may pose an increasing threat in a several ways to the surrounding natural habitats in the future.

In the first instance, because of the contributing factors of the nutrient enrichment due to forage, the increased amount of urine and waste, the persistent weed seeds introduced with contaminated feed, digging and trampling due to a higher concentration of animals (which has been proven in previous research by Rusvai et al., 2022) [12,13], the effects of seed dispersal mechanisms [47], and the above-mentioned climatic factors, bait sites can even be the focal points of biological invasions. Considering that there is a very large number of bait sites in the country (about 30,000), and they are used regularly and very intensively, they can also serve as major infection hotspots in a network. Thus, although weed infestation typically remains local, and the light-demanding weed species that become abundant in small clearings are presumably not expected to spread to neighboring forest areas [48], their effect may occur even farther away in the case of suitable habitat patches (e.g., disturbed clearings, open forest patches, and unclosed regeneration patches) by spreading via the road network and in other ways [49].

All in all, it can be said that the least environmental damage would be achieved by the prohibition or significant limitation of bait sites in nature conservation areas. It would also be worthwhile to plan more comprehensive studies that focus on the role of bait sites in changes at the local and landscape level as well, regarding the spread and establishment of invasive and weed species.

**Author Contributions:** Conceptualization, J.H., K.R. and S.C.; methodology, K.R.; software, J.H.; validation, K.R., J.H. and S.C.; formal analysis, K.R.; investigation, K.R.; resources, K.R.; data curation, K.R. and J.H.; writing—original draft preparation, K.R.; writing—review and editing, J.H. and S.C.; visualization, K.R.; supervision, J.H. and S.C. All authors have read and agreed to the published version of the manuscript.

**Funding:** This research received no external funding.

**Data Availability Statement:** The data presented in this study are available on request from the first author.

**Acknowledgments:** We sincerely thank the editors and the anonymous reviewers for their insightful comments and suggestions.

**Conflicts of Interest:** The authors declare no conflicts of interest.

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
