# Peer review of "The Effect of Climate on Strongly Disturbed Vegetation of Bait Sites in a Central European Lower Montane Zone, Hungary"

_land, doi:10.3390/land13071108_

Round 1

Reviewer 1 Report (Previous Reviewer 3)

Comments and Suggestions for Authors

The authors took all my comments seriously and used their professional judgment to incorporate them to improve the English in the article. This was a signficant revision that required and demonstrated considerable care. I also think that the revision improved the presentation of the significance of the study. This is a sound and innovative research study that definitely deserves publication. 

Reviewer 2 Report (Previous Reviewer 5)

Comments and Suggestions for Authors

Nothing got improved in the revised version. Still the same weekness. Authors did not answer or improve the critics that was raised and just said in the future, we will improve it, which is not acceptable and I must reject the manuscript.

Comments on the Quality of English Language

Still so poor quality

This manuscript is a resubmission of an earlier submission. The following is a list of the peer review reports and author responses from that submission.

Round 1

Reviewer 1 Report

Comments and Suggestions for Authors

The human landscape transforming activity contributes to the global change of vegetation in different forms. Hunting is one of the most ancient human landscape-shaping activities. Feeders for hunting are particularly disruptive to vegetation. In the present study, was conducted a vegetation survey in these highly disturbed places. Was investigated the vegetation dynamics over several years in in the turkey oak–sessile oak zone, in two areas with different moisture and shade conditions (forest and clearing). One important background factor is changes in precipitation and temperature.

The present study is interesting and provides a detailed analysis of the effect of climate on the regeneration capacity of strongly disturbed vegetation in a Central European Lower Montane Zone (Hungary).

The knowledge produced by this paper may be of considerable interest to a wide audience, both for the methodology of application of the project ideas and for providing precise guidelines and insights to sector specialists.

The novelty of the present study lies in the fact that the observation takes place with continuous disturbance, where the human influence is very large. The results confirm that the weed infestation be detectable at bait sites over a long period, and the interannual changes were typical in every year, according to the fact that in August, more weed species are usually present with a greater cover at the bait sites. The main meteorological factors played a role in the degree of weed infestation in each year. There was a higher proportion of vegetation regeneration in wetter years, while dry periods do not favor successional processes. With the drying of the climate, the disturbed areas are constantly losing their natural value, but wetter weather is not an automatic solution either. A more precise understanding of vulnerable systems and careful planning of treatment interventions is necessary.

A more precise understanding of vulnerable systems and careful planning of treatment interventions is necessary. This publication establishes a foundation for further research.

I therefore suggest its publication precisely in view of the need to proceed with concrete actions concerning a basis for detailed analysis of the effect of climate on the regeneration capacity of strongly disturbed vegetation.

Author Response

Detailed response to Referees:

Referee’s comment are with black letters, our responses in italics and in blue colors.

Reviewer 01

Points should be improved:

  • Does the introduction provide sufficient background and include all relevant references?
  • Are the results clearly presented?
  • Are the conclusions supported by the results?

The human landscape transforming activity contributes to the global change of vegetation in different forms. Hunting is one of the most ancient human landscape-shaping activities. Feeders for hunting are particularly disruptive to vegetation. In the present study, was conducted a vegetation survey in these highly disturbed places. Was investigated the vegetation dynamics over several years in in the turkey oak–sessile oak zone, in two areas with different moisture and shade conditions (forest and clearing). One important background factor is changes in precipitation and temperature.

The present study is interesting and provides a detailed analysis of the effect of climate on the regeneration capacity of strongly disturbed vegetation in a Central European Lower Montane Zone (Hungary).

The knowledge produced by this paper may be of considerable interest to a wide audience, both for the methodology of application of the project ideas and for providing precise guidelines and insights to sector specialists.

The novelty of the present study lies in the fact that the observation takes place with continuous disturbance, where the human influence is very large. The results confirm that the weed infestation be detectable at bait sites over a long period, and the interannual changes were typical in every year, according to the fact that in August, more weed species are usually present with a greater cover at the bait sites. The main meteorological factors played a role in the degree of weed infestation in each year. There was a higher proportion of vegetation regeneration in wetter years, while dry periods do not favor successional processes. With the drying of the climate, the disturbed areas are constantly losing their natural value, but wetter weather is not an automatic solution either. A more precise understanding of vulnerable systems and careful planning of treatment interventions is necessary.

This publication establishes a foundation for further research.

I therefore suggest its publication precisely in view of the need to proceed with concrete actions concerning a basis for detailed analysis of the effect of climate on the regeneration capacity of strongly disturbed vegetation.

>>> Answer:

Thank you for reviewing our manuscript and we are very grateful that you recommend the manuscript for publication. Based on the other reviewer's suggestions the quality of English in the manuscript was improved. We also made corrections in Results and Conclusions to make our results more understandable.

Reviewer 2 Report

Comments and Suggestions for Authors

The manuscript entitled ‘The effect of climate on the regeneration capacity of strongly disturbed vegetation in a Central European Lower Montane Zone, Hungary’ requires a through rewriting, revisions and incorporation of the suggested changes. The manuscript is not yet suitable for acceptance in its current state. The authors are asked to resubmit a thoroughly revised version, with acceptance or rejection contingent upon the implementation of the suggested changes.

Please find attached the accompanying review comments for the reference.

Comments on the Quality of English Language

The English language quality is generally good and acceptable. However, there is a scope of improvement and the authors are asked to avoid constructing long sentences that impede proper understanding of the concepts.

Author Response

Reviewer 02

Points should be improved:

Are all the cited references relevant to the research?

The manuscript entitled ‘The effect of climate on the regeneration capacity of strongly disturbed vegetation in a Central European Lower Montane Zone, Hungary’ requires a through rewriting, revisions and incorporation of the suggested changes. The manuscript is not yet suitable for acceptance in its current state. The authors are asked to resubmit a thoroughly revised version, with acceptance or rejection contingent upon the implementation of the suggested changes. Please find attached the accompanying review comments for the reference.

The English language quality is generally good and acceptable. However, there is a scope of improvement and the authors are asked to avoid constructing long sentences that impede proper understanding of the concepts.

The manuscript entitled ‘The effect of climate on the regeneration capacity of strongly disturbed vegetation in a Central European Lower Montane Zone, Hungary’ requires a through rewriting, revisions and incorporation of the suggested changes. The manuscript is not yet suitable for acceptance in its current state. The authors are asked to resubmit a thoroughly revised version, with acceptance or rejection contingent upon the implementation of the suggested changes.

General comments

− The title does not align with the experiment's objectives. It includes the term 'regeneration capacity,' which is a broad and significant term, yet this concept is not thoroughly explored in the study. The authors are advised to avoid using terms that do not directly relate to the current scope of the experiment.

>>> We are grateful for the suggestion.  We changed the title: “The effect of climate on strongly disturbed vegetation of bait sites in a Central European Lower Montane Zone, Hungary”

− The manuscript presents numerous concepts without adequate experimental evidence and explanations. It seems to rely heavily on visual interpretation rather than providing robust scientific and statistical justifications. While many concepts are introduced, they lack organization and thorough presentation and discussion within the manuscript. Accordingly, the authors are encouraged to enhance the clarity of the manuscript, particularly within the methodology and discussion section.

>>> We are grateful for the suggestion. Following the special comments, we made changes in Results and Conclusions sections to improve the structure of the manuscript and the interpretation of the results.

− Using degradation indicator species to assess soil degradation is one approach, but it's not exhaustive. Soil degradation involves multiple factors like erosion, loss of organic matter, compaction, contamination and changes in structure and chemistry. While indicator species provide insight, including other indicators such as soil properties, land use history and landscape characteristics would give a fuller picture. Moreover, the exclusive reliance on temperature and precipitation data in the experiment may not encompass all factors influencing soil degradation, as also indicated in the manuscript (Line 258-259). Accordingly, the authors are requested to provide justification for the selected approach.

>>> Thank you very much for the reviewer's suggestion .We made an addition to 2.2. chapter to understand why we used this (Borhidi Social Behaviour Type (SBT) classification) grouping system and what the individual categories are based on: “This is an adapted version of Ellenberg’s grouping (Ellenberg et al., 1991) and Grime’s competitor/stress-tolerator/ruderal (CSR) plant functional type system (Grime, 1979) to Pannonian flora. This grouping derives from species behavior and ecological attributes at a given observation level (Borhidi, 1995). Social behavior can be defined as the role that a plant species plays in the community considering its ecological characteristics, morphology and physiological performance.”

Specific comments

Abstract

− The abstract lacks clarity, particularly in defining the problem statement and presenting the exact findings of the experiment. The authors are advised to include both quantitative details and descriptive information to enhance clarity.

>>> Thank you very much for the reviewer's suggestion. We rewrited the abstract: “Abstract: The human landscape transforming activity contributes to the global change of vegetation in different forms. Hunting is one of the most ancient human landscape-shaping activities. Feeders for hunting are particularly disruptive to vegetation. In the present study, we conducted a vegetation survey in these highly disturbed places. We are investigated the vegetation dynamics over several years in the turkey oak–sessile oak zone, in two areas with different moisture and shade conditions (forest and clearing). One important background factor is changes in precipitation and temperature. Our results confirm that weed infestation is detectable at bait sites over a long period. The seasonal changes in field weed vegetation, as well as the increase in the number and cover of weed species at the end of summer, resulting from lifestyle characteristics, were generally detectable in all years and locations, especially in the case of open and more strongly degraded clearings. The main meteorological factors played a role in the degree of weed infestation in each year. Degradation was more significant in drought years, while regeneration was also observed in wetter periods. At baits located in clearing, we showed a positive correlation between the amount of summer precipitation and the total cover of weed species, as well as between the average spring temperature and the cover of certain weed species. With the drying of the climate, the disturbed areas are constantly losing their natural value, but wetter weather is not an automatic solution either. Considering, that there are approx. 30,000 bait sites in the country, and they are used reg-ularly and very intensive, they can serve as major infection hotspots for alien species in a network.”

Introduction

− Line 31-75: The introduction begins abruptly, lacking a discussion of the context and relevance to the present experiment before delving into the experimental setup. Furthermore, the introduction is too brief to adequately justify and cover all the concepts that require elaboration. Additionally, the authors are asked to avoid constructing long sentences that impede proper understanding of the concepts.

>>> Thank you very much for the reviewer's suggestion. We put a paragraph before it: “Feeding wildlife is a controversial method with diverse effects on ecosystems, animal welfare and public safety. Despite the level of baiting and the supplementary feeding of wild game, our understanding of the ecological effects of these types of food support is still very limited (Selva et al., 2014). Artificial feeding may have direct benefits, like reducing mortality, enhancing the physical wellbeing of game, maintaining body weight and re-productive performance (Richardson, 2006). However, it can lead to severe problems, like changes in social and territorial behaviour, the natural migration and activity patterns of animal species, it can cause an increase in competitiveness, and lead to negative interactions between animals and increased risk of disease transmission (Milner et al., 2014; Sorensen et al., 2014).”

− Line 60-64: “Others found that some exotic species…..Therefore, it can be assumed that climate change…..at these places.” Please provide a suitable explanation for how the species found in higher altitudes may be linked to climate change, and ensure that the concepts and the sentences are appropriately connected.

>>> Thank you very much, we changed to: “Others found that some exotic species can be detectable even in the mountainous environment at higher and higher altitudes, even above 1000 m near these sites (Kochjarová et al. 2023). Therefore, it may be that climate change can also affect the presence and the abundance of weed species at the feeding grounds…”

− Line 68-69: “The most important… temperature”. This sentence lacks relevance when discussing the specific objectives of the experiment. It seems more like a factor that might have influenced the experiment's direction. The authors are advised to remove this sentence.

>>> Thank you very much, we removed this sentence.

− Line 71-76: The objectives lack direction and need to be presented in a more scientific manner.

>>> Thank you very much, we changed the questions of the study:

How does the extent of weed cover change over the years and whether seasonal changes be detected?

How does precipitation and temperature affect the number and the cover of weed species in each year?

Materials and Methods

− Line 80: Please incorporate the geographic coordinates (latitude and longitude) for both the study area and the sampling sites. This can be achieved by either revising the map (Figure 1) to include coordinates or by providing the details within the manuscript text.

>>> Thank you very much, we have supplemented the map.

− Line 83: “All bait sites… 5–10 years)”. Kindly present the criteria used for selecting the bait sites and describe how the assessment was conducted.

>>> We reworded that part: Since there is no official public register of the bait sites available, we have chosen those that have certainly been in use for at least 5-10 years and are located in forest or clearing (not in edge habitat).

− Line 86-87: Please include a diagram illustrating the investigation procedure, integrating it with Figure 1.

>>> We are grateful for the suggestion , we also made changes in Figure 1 and rewrites the title of the figure as well.

− Line 88: The authors need to explain why the vegetation cover assessments were conducted in May and August of each year. Ideally, a comprehensive vegetation cover assessment over the course of a year should be accompanied by a well-thought-out sampling strategy, considering assessments in both pre- and post-monsoon seasons. In Line 68, it seems the authors aimed to highlight the influence of precipitation and temperature on weed infestation levels. Therefore, it's important to justify the timing of the assessments and the rationale behind their selection.

>>> We added a sentence to explain why the assessments were conducted in May and August: Periods were chosen in order to examine whether the well-known interannual changes of the arable fields (Pinke et al. 2012) can also be detected at bait sites.

− Line 97: The distinction between the forest bait site and clearing bait site is unclear in sub-section 2.1, leading to confusion. The authors are urged to provide a clear description of these sites within the context of the current experiment.

>>> We are grateful for the suggestion, we added supplementary information: “Since there is no official public register of the bait sites available, we have chosen those that have certainly been in use for at least 5-10 years and are located in forest or clearing (not in edge habitat).

Forest sites are situated in oak stands, where the canopy cover was at least 80% and the forest floor was characterized by dense litter cover and sparse undergrowth. Clearing sites refer to small patches (50–100 m in diameter) of dry-mesophilous mountain meadows, which were created by tree cutting several decades ago. The selected sites are located on flat-topped ridges or saddle areas with gently sloping and slight southern exposure at an altitude of 400-500 m above sea level.”

− Line 99: The authors are asked to provide details whenever introducing a new term or concept initially. The term ‘naturalness of communities’ should be adequately introduced before further exploration. Additionally, the classification of weed species should be thoroughly discussed, considering its significant role as a criterion in the present assessment.

>>> We added plus sentences: “The naturalness of the communities was evaluated based on the proportion of these groups. The ratio of the species groups, as well as their proportion of the cumulative cover was also calculated.”

− Line 115: For the experiment, the Tukey HSD test was employed for pairwise comparisons. Could you specify the groups or factors compared in this analysis?

>>> We added plus sentences : For post hoc test the Tukey Honestly Significant Difference (HSD) with corrections (adjusted p-values for the multiple tests) was used. The groups were the individual places and years, as can be seen in this detail. Among the results, only the significant differences were shown.

$Y

                   diff        lwr          upr     p adj

Cle18-Cle16  -0.5000000  -8.978933   7.97893347 0.9999995

Cle19-Cle16   6.1666667  -2.312267  14.64560014 0.3059658

Cle20-Cle16   6.0000000  -2.478933  14.47893347 0.3389583

− Line 118: What is the rationale behind employing linear regression? Additionally, could you clarify the nature of the PAST program? It's not evident from the current explanation whether PAST is a statistical tool or an algorithm utilized for establishing linear regression. Please revise this for clarity accordingly.

>>> With the help of linear regression, we wanted to explore the effect of precipitation and temperature experienced in each year..We made an addition: For linear regression we used the PAST program [Hiba! A hivatkozási forrás nem található.]. We used a comprehensive yet easy-to-use software package long used in paleontology to perform quantitative standard numerical analyzes and operations. Most of the most frequent functions used in PAST (PAleontological Statistics) can be used in paleontology as well as in ecology, and are suitable for handling often incomplete data. These functions are not found in standard, more extensive statistical packages, but at the same time they can be used well in university education.

Results

− Line 121-122: The authors should clarify whether they discovered and categorized the species themselves or if the information was obtained from other literature sources. If the species details are original findings, they should consider including them in the appendix or supplementary materials.

>>> We made an addition: “This is an adapted version of Ellenberg’s grouping (Ellenberg et al., 1991) and Grime’s competitor/stress-tolerator/ruderal (CSR) plant functional type system (Grime, 1979) to Pannonian flora. This grouping derives from species behavior and ecological attributes at a given observation level (Borhidi, 1995). Social behavior can be defined as the role that a plant species plays in the community considering its ecological characteristics, morphol-ogy and physiological performance. According to this grouping, two main groups were used:….”

− Line 207: How many degradation indicator species were included in the assessment for constructing the scatterplots shown in Figure 5? Additionally, the authors are requested to provide a descriptive justification of the relationship based on the R2 values.

>>>Thank you very much for the suggestion, the scatter plot was really not appropriate, we deleted it from the text.

In our case, the R2 value is R=0.975 and R=0.831, that is, it approaches 1, that is, the estimation of the coefficients in the so-called linear model will be closely related.

− Line 216-219: How does the significant impact of summer precipitation on weed growth rate relate to degradation? The authors are asked to explain this phenomenon in relation to the degradation indicator species. Additionally, please clarify how this assessment aligns with the current objectives of the study.

>>> We made additions to the discussion: “warm, dry springs and the following rainy summers increased weed density. The reason for this is that large weed species usually germinate later (Berzsenyi 2000), and they require significant amount of water for their growth (Lehoczky et al. 2016), thus they are able to suppress natural species.”

− Line 244-246: The authors are asked to provide justification for the provided data and to clarify the methodology used for calculations.

>>> We changed the sentence: “The proportion of these species from the cumulative weed coverage varied between 2.2% to 46.5% during the study periods.” And also made an addition in the material and methods: “The naturalness of the communities was evaluated based on the proportion of these groups. The ratio of the species groups, as well as their proportion of the cumulative cover was also calculated.”

− Line 251: In Table 3, the authors are requested to provide the explanation of the methodology used for calculations. Additionally, what is the statistical accuracy of this interpretation? >>> We added a footnote to the table: “*Values show percentage increase in August compared to May, based on the cumulative cover of the listed species groups.”

− Line 253: What distinguishes ‘alien weeds’ from ‘invasive alien plant species’? The authors are asked to provide justification for any differences identified. If there are no distinctions, the authors are encouraged to use a consistent nomenclature to describe these entities.

>>> Aliens weeds are only alien but not invasive. We changed alien weed to alien plant.

Discussion

− Line 312-314: Meteorological factors such as temperature and rainfall could have influenced the dominance of alien weed species. However, the data to capture this variation is not adequately presented in terms of statistical interpretation compared to other weed species. Additionally, the Naturalness indicator species listed in Table 2 may employ dormancy mechanisms, ensuring their longevity in the soil seed bank until favourable conditions that trigger germination. Therefore, it raises the question of why only meteorological factors such as temperature and rainfall affected the alien weed species and not the native naturalness indicator species. Please provide justification for this assertion.

>>> We added a sentence to explain this: “Because although the disturbance is slightly reduced, the nutrient content of the soils was still very high (Rusvai et al. 2022b), which mainly helps the growth of the species.”

Conclusions

− Line 352-394: The conclusion section should align with the experimental findings of the current investigation, focusing on the study's results rather than referencing other studies. Relevant discussions and justifications may be relocated to the introduction section or throughout the manuscript, rather than being included solely in the discussion section. The authors are advised to re-write the conclusion section to accurately reflect the specific findings of the present experiment.

>>> We shortened the conclusion chapter and moved some parts to the discussion.

Reviewer 3 Report

Comments and Suggestions for Authors

Title: I think the title does not quite inform the reader what the content is. Is there a way to mention something about bait sites, hunting wild boars, or potential adverse effects on native forests from the many bait sites? I think the phrase “the regeneration capacity” is not very informative and could be left out and “strongly disturbed vegetation” is a little vague. Could work with omitting or changing these two phrases to make it possible to include something about these other matters. Not essential, but just trying to be sure that the significance about a larger issue is made clearer. It was surprising to me to hear that there are 30,000 bait sites, which make it seem that this is a very serious threat. But, words that highlight that threat are not in the title. 

I am a native English speaker from the USA. The English in the article is quite good, but there are edits that I think could improve it a little. However, my many suggestions may differ from how English is spoken in your area, so use your judgment.

Line 16. are investigating the COULD BE investigated

Line 17. No need for two “in in” 

Line 21. be detectable COULD BE is detectable

Lines 27-28. “A more precise understanding...” sentence could be something more firm about this article’s importance. I would omit this sentence and instead mention something from the sentence on Lines 370-372, which makes it clear how important your findings are.

Keywords maybe should include “wild boar hunting” or something like this.

Line 32. I think it is very important to have the first sentence highlight the importance of the issue and your study. This should mention, I think, the 30,000 estimate, explain briefly that there is a large threat from these bait sites and what your contribution is. This is what can be called a “hook” that aims to interest the reader right away and make sure they understand what your study contributes. This makes it more likely that readers will look more closely at your findings. Great to do something similar as the first sentence of the Conclusions too. Hook the reader in both places, like you would catch a fish with some bait! Don’t overdo it, but get it out there clearly and briefly. 

Line 38. feed generally COULD BE feed was generally

Line 42. Delete “cause” or “lead to” as they mean something similar

Line 55. I don’t know what “segetal” is, but perhaps this is generally known in your study area

Line 61. Change “along” to “near”

Line 62. Change “can be assumed” to “may be”

Line 82. Please include Latin names for “turkey oak-sessile oak”

Line 87. I don’t know what “22-22 1x1 meter” means. Is that correct?

Line 91. Semicolon between parameters and average

Line 92. Change “was used” to “were used” and Change “Data was” to “Data were” as data is plural, datum is the singular version.

Line 101. Change “group was” to “groups were”

Line 106. Omit “with”

Line 107. Change “it” to “them”

Line 117. Add “the” before “PAST”

Line 121. Change “years” to “year”

Line 127. Change “weed” to “weeds”

Line 150. Bromus sterilis should be in italics

Figure 2. Please label the y-axis and explain the units. Why are there 4 of Cle and 4 of For when Figure 1 shows only three of each? The label “Y” for the x-axis is unclear; why does it not just say something like “Sampling units”?

Figure 3. Same comments as Figure 2.

Figure 4. The text for the axis labels and the numbers above the bars could be bigger so they are more readable. To me it is not clear that there is a trend for Clearing. You can do a formal statistical trend test if you have access to a statistical analysis program.

Line 203. “..” should just be one: “.”

Figure 5. What are the units for the Y-axis? Total cover usually is in percent, so this must be something else? Is it CÌŠ or ÌŠC? Here again it appears there are four sites, but the map has only three sites.

Figure 6. Need to color spring and summer precipitation so they are distinct. I suggest blue for spring, since it is cooler and red for summer since it is warmer. I don’t understand how the Y-axis (Totalcover) can be > 100.0%. Please explain this somewhere and also in the Figures. I may have missed your explanation if it is there already. Also change “Totalcover” to “Total Cover” I think the X-axis label should be “Year” not “Time” Also, the caption says “and average temperature” but I do not see any temperature line or values. Also, what does “1-5. quadrats” mean?–that does not make sense to me.

Line 245. Change “plant” to “plants”

Line 251. Change “baits” to “bait”

Line 253. What is the source for identifying species as IAS? Perhaps cite Lanbdon et al. earlier.

Lines 267-268. “the assumption that was established investigating the impact of hunting” is awkward. I don’t know how to rewrite it. 

Line 269. Perhaps needs a period after “here” and then capitalize “Thanks”

Line 276. “year” should be “years”

Lines 277-278. I would change this to: “...bait; further away from the center their density and the number of species decreased...”

Line 279. “types” should be “type”

Line 282. “with” should be “by”

Line 284. “phenomena presumably” should be “phenomena are presumably”

Line 286. “was” should be “were”

Line 288. “like” should be “as”; also, the three Latin names should be in italics

Line 289. “was” should be “were”

Line 304. “environment” should be “environments”

Line 307. It seems that “(see in detail in the next chapter)” should be omitted.

Line 330. “than native plant” should be “than do native plants”

Lines 353-354. I think “forest-clearing” would be clearer as “forest versus clearing”

Line 354. “has” should be “have”

Line 356. Only need one “species”

Lines 370-372. Wow, 30,000 bait sites. Please put this whole sentence or something like it into the very early part of the paper, as it shows how significant your study is. You can repeat this number here too in conclusions. 

In Conclusions, I was told by a reviewer of a paper I published in an MDPI journal that MDPI does not favor including citations in Conclusions, but I am not sure that this is the journal’s policy. It could have been just a reviewer’s desire. It makes some sense, though, to do it this way. Also, you have quite a bit of material that I think is important to have in the Discussion, as it is new material that is not in the Discussion. I think it would be better to move new material to the Discussion, do the citations there, and then summarize the results and discussion in the Conclusions without citations. You can still include the important points that you need to make in the Conclusions, but not have to cite any papers. Also, I think the Conclusions could be a little shorter so that the important findings are highlighted.  

Lines 391-392. I would omit “In terms of scientific part” as this implies that your other Conclusions are not scientific. 

Comments on the Quality of English Language

See above

Author Response

Reviewer 03

Minor editing of English language required.

Points should be improved:

  • Does the introduction provide sufficient background and include all relevant references?
  • Are the results clearly presented?
  • Are the conclusions supported by the results?

Title: I think the title does not quite inform the reader what the content is. Is there a way to mention something about bait sites, hunting wild boars, or potential adverse effects on native forests from the many bait sites? I think the phrase “the regeneration capacity” is not very informative and could be left out and “strongly disturbed vegetation” is a little vague. Could work with omitting or changing these two phrases to make it possible to include something about these other matters. Not essential, but just trying to be sure that the significance about a larger issue is made clearer. It was surprising to me to hear that there are 30,000 bait sites, which make it seem that this is a very serious threat. But, words that highlight that threat are not in the title.

We are grateful to Referee 2 forthe  positive general  evaluation and for  detailed and constructive suggestions. These comments helped us a lot to improve our MS. We respond point to point to these comments below.

>>>We changed the title: The effect of climate on strongly disturbed vegetation of bait sites in a Central European Lower Montane Zone, Hungary

I am a native English speaker from the USA. The English in the article is quite good, but there are edits that I think could improve it a little. However, my many suggestions may differ from how English is spoken in your area, so use your judgment.

- Line 16. are investigating the COULD BE investigated

- Line 17. No need for two “in in” 

- Line 21. be detectable COULD BE is detectable

- Lines 27-28. “A more precise understanding...” sentence could be something more firm about this article’s importance. I would omit this sentence and instead mention something from the sentence on Lines 370-372, which makes it clear how important your findings are.

- Keywords maybe should include “wild boar hunting” or something like this.

- Line 32. I think it is very important to have the first sentence highlight the importance of the issue and your study. This should mention, I think, the 30,000 estimate, explain briefly that there is a large threat from these bait sites and what your contribution is. This is what can be called a “hook” that aims to interest the reader right away and make sure they understand what your study contributes. This makes it more likely that readers will look more closely at your findings. Great to do something similar as the first sentence of the Conclusions too. Hook the reader in both places, like you would catch a fish with some bait! Don’t overdo it, but get it out there clearly and briefly.

>>> We placed this paragraph to Introduction: “In Hungary, bait sites for wild boar hinting are the most widespread, and the problem is not only their large number (according to estimates, around 30,000 are in operation across the country), but also their extremely intensive use. According to the reported data alone, an average of nearly 150,000 tons of fodder is dumped on these sites every year, which significantly endangers the natural habitats due to fodder contaminated with weed seeds and constant vigorous disturbance.”

- Line 38. feed generally COULD BE feed was generally

We accepted this suggestion. Please, see in the revised text.

- Line 42. Delete “cause” or “lead to” as they mean something similar We accepted this suggestion

- Line 55. I don’t know what “segetal” is, but perhaps this is generally known in your study area > arable weed We accepted this suggestion

- Line 61. Change “along” to “near” We accepted this suggestion

- Line 62. Change “can be assumed” to “may be” We accepted this suggestion

- Line 82. Please include Latin names for “turkey oak-sessile oak” We accepted this suggestion

- Line 87. I don’t know what “22-22 1x1 meter” means. Is that correct? > 22 tangential quadrats (1x1m) were placed on each of them We accepted this suggestion

- Line 91. Semicolon between parameters and average We accepted this suggestion

- Line 92. Change “was used” to “were used” and Change “Data was” to “Data were” as data is plural, datum is the singular version. We accepted this suggestion

- Line 101. Change “group was” to “groups were” We accepted this suggestion

- Line 106. Omit “with” > we calculated with monthly averages We accepted this suggestion

- Line 107. Change “it” to “them” We accepted this suggestion

- Line 117. Add “the” before “PAST” We accepted this suggestion. Please, see in the revised text.

- Line 121. Change “years” to “year” > in the 4-year study period. We accepted this suggestion. Please, see in the revised text.

- Line 127. Change “weed” to “weeds” We accepted this suggestion. Please, see in the revised text.

- Line 150. Bromus sterilis should be in italics We accepted this suggestion. Please, see in the revised text.

- Figure 2. Please label the y-axis and explain the units. Why are there 4 of Cle and 4 of For when Figure 1 shows only three of each? The label “Y” for the x-axis is unclear; why does it not just say something like “Sampling units”?

Thank you, we accepted this suggestion and modified the Figure.

- Figure 3. Same comments as Figure 2.

Thank you, we accepted this suggestion and modified the Figure.

- Figure 4. The text for the axis labels and the numbers above the bars could be bigger so they are more readable. To me it is not clear that there is a trend for Clearing. You can do a formal statistical trend test if you have access to a statistical analysis program.

>>> Thank you for suggestion, the text for the axis labels and the numbers made bigger. Thank you for the suggestion, in this case we can only talk about seasonal fluctuation.

- Line 203. “..” should just be one: “.”

- Figure 5. What are the units for the Y-axis? Total cover usually is in percent, so this must be something else? Is it CÌŠ or ÌŠC? Here again it appears there are four sites, but the map has only three sites.

>>>Thank you very much for the suggestion, in this case we are dealing with four examination years, the places have been treated together here. The unit of the species number is the number of pieces, we cannot indicate this otherwise.

- Figure 6. Need to color spring and summer precipitation so they are distinct. I suggest blue for spring, since it is cooler and red for summer since it is warmer. I don’t understand how the Y-axis (Totalcover) can be > 100.0%. Please explain this somewhere and also in the Figures. I may have missed your explanation if it is there already. Also change “Totalcover” to “Total Cover” I think the X-axis label should be “Year” not “Time” Also, the caption says “and average temperature” but I do not see any temperature line or values. Also, what does “1-5. quadrats” mean?–that does not make sense to me.

>>> Thank you very much for the suggestion. The figure has been corrected as suggested.Due to the structure of the vegetation, it has several levels, so the areas covered by each species can overlap. Since the total coverage is the percentage coverage of the species present in the sampling square, the coverage can be more than 100%.

- Line 245. Change “plant” to “plants” We accepted this suggestion

- Line 251. Change “baits” to “bait” We accepted this suggestion

- Line 253. What is the source for identifying species as IAS? Perhaps cite Lanbdon et al. earlier.

>>> We cited Lambdon et al.: “Although, invasive alien plant species (IAS; according to Lambdon et al. 2008) usually accounted for…”

- Lines 267-268. “the assumption that was established investigating the impact of hunting” is awkward. I don’t know how to rewrite it.

>>> We rewrite it: “Our results are compatible with those experienced in Slovakia, in connection with hunting establishments.”

- Line 269. Perhaps needs a period after “here” and then capitalize “Thanks” We accepted this suggestion

- Line 276. “year” should be “years” We accepted this suggestion

- Lines 277-278. I would change this to: “...bait; further away from the center their density and the number of species decreased...” We accepted this suggestion

- Line 279. “types” should be “type” We accepted this suggestion

- Line 282. “with” should be “by” We accepted this suggestion

- Line 284. “phenomena presumably” should be “phenomena are presumably” We accepted this suggestion

- Line 286. “was” should be “were” We accepted this suggestion

- Line 288. “like” should be “as”; also, the three Latin names should be in italics We accepted this suggestion

- Line 289. “was” should be “were” We accepted this suggestion

- Line 304. “environment” should be “environments” We accepted this suggestion

- Line 307. It seems that “(see in detail in the next chapter)” should be omitted. We accepted this suggestion

- Line 330. “than native plant” should be “than do native plants” We accepted this suggestion

- Lines 353-354. I think “forest-clearing” would be clearer as “forest versus clearing” We accepted this suggestion

- Line 354. “has” should be “have” We accepted this suggestion

- Line 356. Only need one “species” We accepted this suggestion

- Lines 370-372. Wow, 30,000 bait sites. Please put this whole sentence or something like it into the very early part of the paper, as it shows how significant your study is. You can repeat this number here too in conclusions.

>>> In conclusions, the high number of the bait sites is mentioned.

- In Conclusions, I was told by a reviewer of a paper I published in an MDPI journal that MDPI does not favor including citations in Conclusions, but I am not sure that this is the journal’s policy. It could have been just a reviewer’s desire. It makes some sense, though, to do it this way. Also, you have quite a bit of material that I think is important to have in the Discussion, as it is new material that is not in the Discussion. I think it would be better to move new material to the Discussion, do the citations there, and then summarize the results and discussion in the Conclusions without citations. You can still include the important points that you need to make in the Conclusions, but not have to cite any papers. Also, I think the Conclusions could be a little shorter so that the important findings are highlighted.

>>> We shortened the conclusion chapter and moved some parts to the discussion.

- Lines 391-392. I would omit “In terms of scientific part” as this implies that your other Conclusions are not scientific. 

Reviewer 4 Report

Comments and Suggestions for Authors

This manuscript presents an interesting study on weed species at bait sites. Weed species at six bait sites in four years were analyzed based on field survey. In general, the topic is valuable for understanding land degradation and recovery processes. However, the new findings and contributions of this study were not clearly illustrated. The results section should be improved, there are many redundant descriptions.

1.      Title. The Title needs to be improved. Can “regeneration capacity“ be linked to “weed species” directly?

2.      Abstract. The abstract should be improved. After reading the abstract, I found it is difficult to capture the main contribution of this study.

3.      Line 87. “22-22 1x1 meter”. Confusing.

4.      Line 89-90. “the coenological data of the individual units are evaluated together, in order to filter out spatial autocorrelation”. It is not clear how the spatial autocorrelation was filtered out.

5.      Line 90-92. What kinds of meteorological data were used? Grid data or data from weather stations? Are the data appropriated for the six sites.?

6.      It is described that the sites are located in different moisture and shade conditions. It is better to show the terrain of the sites to support the description. Elevation, slope and aspect is important for species distribution.

7.      Line 114-115. “The effects of different sites were tested by repeated–measures analysis of variance (ANOVA)”. What kinds of effects? Confusing.

8.      Tables 1 and 2 show the weed species at all forest and clearing sites. Are there differences within the forest sites? Are there differences within the clearing sites. It is important to show the differences within a site type.

9.      Tables 1 and 2. It is better to explain the abbreviations in the table captions.

10.   Line 163-165. No Figure supports the spatial pattern of species within the sites.

11.   Line 204. “Influence of meteorological factors” on What?

12.   Figure 5. There are only four points for linear regression. hence, it may be statistically meaningless.

13.   Line 214-224. There is some inconformity between the text and figure 6. In addition, It is better to avoid citation in the results.

14.   Table 3? Confusing on the data. 790% and 1763%. How were these data calculated?

15.   Conclusion should be improved. There are many citations. In this section, the content should be focused on your new findings and contribution.

Author Response

Reviewer 04

Points should be improved:

  • Does the introduction provide sufficient background and include all relevant references?
  • Is the research design appropriate?

This manuscript presents an interesting study on weed species at bait sites. Weed species at six bait sites in four years were analyzed based on field survey. In general, the topic is valuable for understanding land degradation and recovery processes. However, the new findings and contributions of this study were not clearly illustrated. The results section should be improved, there are many redundant descriptions.

  1. Title. The Title needs to be improved. Can “regeneration capacity“ be linked to “weed species” directly? >>> The title was changed to ‘The effect of climate on strongly disturbed vegetation of bait sites in a Central European Lower Montane Zone, Hungary’

  1. Abstract. The abstract should be improved. After reading the abstract, I found it is difficult to capture the main contribution of this study. >>> We rewrote the abstract.
  2. Line 87. “22-22 1x1 meter”. Confusing. >>> We modified it: “22 tangential quadrats (1x1m)”
  3. Line 89-90. “the coenological data of the individual units are evaluated together, in order to filter out spatial autocorrelation”. It is not clear how the spatial autocorrelation was filtered out.

>>>Thank you for your comment and we agree with it. Here, we were thinking of the 1x1 meter squares that are in contact in space, but in the present study the data were analyzed together, and in a later analysis we will treat the individual parts separately. We deleted this sentence.

  1. Line 90-92. What kinds of meteorological data were used? Grid data or data from weather stations? Are the data appropriated for the six sites.? >>> We used data from weather stations. Yearly and monthly temperature and precipitation.
  2. It is described that the sites are located in different moisture and shade conditions. It is better to show the terrain of the sites to support the description. Elevation, slope and aspect is important for species distribution. >>> We added this sentence to the description: “The selected sites are located on flat-topped ridges or saddle areas with gently sloping and slight southern exposure at an altitude of 400-500 m above sea level.”
  3. Line 114-115. “The effects of different sites were tested by repeated–measures analysis of variance (ANOVA)”. What kinds of effects? Confusing.

>>> We added this sentence to the description. The effects of different vegetation environment of different sites, and the consecutive years on total cover, number of species of different indicators, were compared using analyses of variance (ANOVA) with sites and years as fixed factors

  1. Tables 1 and 2 show the weed species at all forest and clearing sites. Are there differences within the forest sites? Are there differences within the clearing sites. It is important to show the differences within a site type.

 >>> There were no significant differences between the sites, forest baits were very similar to each other, and it was also true for the three clearing sites as well. We made additions to make this clear (“The naturalness indicator species proved to be more abundant at all forest sites” and “At clearing sites, the species pool and density were completely different (Table 2.). Ruderal and segetal weeds proved to be the most abundant at all three locations…”)

  1. Tables 1 and 2. It is better to explain the abbreviations in the table captions.

>>> We added notes to the tables.

  1. Line 163-165. No Figure supports the spatial pattern of species within the sites.

>>> We have referred to a previous article that includes this.

  1. Line 204. “Influence of meteorological factors” on What? >>> We changed the title to ‘Influence of meteorological factors on vegetation’.
  2. Figure 5. There are only four points for linear regression. hence, it may be statistically meaningless.

>>>Thank you for your comment and we agree with it. We definitely want to continue the investigation in the coming years, for now we have drawn conclusions from the amount of data available to us. However, part of the trend is still visible, we will supplement this with new data later on.

  1. Line 214-224. There is some inconformity between the text and figure 6. In addition, It is better to avoid citation in the results. >>> We clarified the text and put the citation to the discussion.
  2. Table 3? Confusing on the data. 790% and 1763%. How were these data calculated? >>> We added a note to the table: “Note: Values show percentage increase in August compared to May, based on the cumulative cover of the listed species groups.”
  3. Conclusion should be improved. There are many citations. In this section, the content should be focused on your new findings and contribution.>> Thank you for this valuable suggestion. We accepted your suggestions, and we improved this section.

Reviewer 5 Report

Comments and Suggestions for Authors

I had the opportunity to review the manuscript “The effect of climate on the regeneration capacity of strongly disturbed vegetation in a Central European Lower Montane Zone, Hungary”. The manuscript studies the vegetation dynamics in two Oak zone with different moisture and shading conditions from 2016 to 2020. This idea is so primitive and nothing new behind the aim of the manuscript. Moreover, the manuscript needs to be re-written in a better way, as it is hard to follow the ideas presented the paper due to the week English and lack of continuity, especially in the introduction. Authors did not pay any attention to the journal guidelines related to the references; I think they were in hurry which is not fine. Finally, the manuscript has 47% plagiarism which is not acceptable.

Introduction: It should be re-written to have a good understanding of the idea. Sentences are so short and did not give the reader the idea behind the manuscript.

The authors need first to discuss why Bait habitat are important, then the problem facing such habitat, e.g. degradation … etc. Afterwards, you should mention why your study is important, as I did not get the importance of such a study. Moreover, you should mention the research question in a better way. Finally, the authors did not mention anything about the naturalized and invasive species in the introduction.

Methods: The vegetation survey should be described clearly and in detail in a good language, now it is not understandable. The statistical analysis is so primitive, using only ANOVA followed by Tukey test is not enough to have conclusive results, try to use Linear Mixed Effect Models.

Results: The results section is not organized at all; it is too missy. Authors should pay attention to present their results based on their research questions.

Discussion: Again, it is not well organized, and avoid repeating results in this section and keep an eye on only discussion your results.

Conclusion: Is too long, it should be concise. A recommendation section should be included.

References: You should follow the journal guidelines as it should be numbered, but now it is too messy.

Comments on the Quality of English Language

The manuscript needs to be re-written in a better way, as it is hard to follow the ideas presented the paper due to the week English and lack of continuity, especially in the introduction

Author Response

Reviewer 05

English very difficult to understand/incomprehensible

I had the opportunity to review the manuscript “The effect of climate on the regeneration capacity of strongly disturbed vegetation in a Central European Lower Montane Zone, Hungary”. The manuscript studies the vegetation dynamics in two Oak zone with different moisture and shading conditions from 2016 to 2020. This idea is so primitive and nothing new behind the aim of the manuscript. Moreover, the manuscript needs to be re-written in a better way, as it is hard to follow the ideas presented the paper due to the week English and lack of continuity, especially in the introduction. Authors did not pay any attention to the journal guidelines related to the references; I think they were in hurry which is not fine. Finally, the manuscript has 47% plagiarism which is not acceptable.

 Introduction: It should be re-written to have a good understanding of the idea. Sentences are so short and did not give the reader the idea behind the manuscript. >>> Thank you for this valuable suggestion. We accepted your suggestions. We rewrote the introduction.

The authors need first to discuss why Bait habitat are important, then the problem facing such habitat, e.g. degradation … etc. Afterwards, you should mention why your study is important, as I did not get the importance of such a study. Moreover, you should mention the research question in a better way. Finally, the authors did not mention anything about the naturalized and invasive species in the introduction.

 Methods: The vegetation survey should be described clearly and in detail in a good language, now it is not understandable.

The statistical analysis is so primitive, using only ANOVA followed by Tukey test is not enough to have conclusive results, try to use Linear Mixed Effect Models.

>>>Thank you very much for your comment and in the future, we will strive to perform statistical analyses as complex as possible. Given that research on the topic is very underrepresented, the research methods are also characterized by pathfinding. We want to understand how the impact factors work, so we started our investigations using the most basic methods.

Results: The results section is not organized at all; it is too missy. Authors should pay attention to present their results based on their research questions. Discussion: Again, it is not well organized, and avoid repeating results in this section and keep an eye on only discussion your results. Conclusion: Is too long, it should be concise.

>>> Thank you for this valuable suggestion. Our research questions have been revised in the Introduction and the whole Results section has been rewritten into new structure and logic. We added also the information you suggested.

A recommendation section should be included.

 >>>Thank you very much for your comment. Recommendations found in the last phrase.

References: You should follow the journal guidelines as it should be numbered, but now it is too messy.

Comments on the Quality of English Language:

The manuscript needs to be re-written in a better way, as it is hard to follow the ideas presented the paper due to the week English and lack of continuity, especially in the introduction

>>> Thank you for this valuable suggestion. We accepted your suggestions. We rewrote several section of MS.